# Diagnostic accuracy of chest ultrasound scan in the diagnosis of childhood tuberculosis

**Geoffrey Erem** [1,2]*, **Caroline Otike**[3], **Maxwell Okuja**[1], **Faith Ameda**[1], **Dorothy Irene Nalyweyiso**[4], **Aloysius Gonzaga Mubuuke**[1], **Michael Kakinda**[3]

1 Department of Radiology, School of Medicine, Makerere University, Kampala, Uganda, 2 Department of Radiology, St Francis Hospital Nsambya, Kampala, Uganda, 3 Directorate of Clinical Services, Joint Clinical Research Center, Kampala, Uganda, 4 Department of Radiology, Mulago National Referral Hospital, Kampala, Uganda

* dreremgeoffrey@gmail.com

**Data Availability Statement:** All relevant data are within the paper and its Supporting Information files.

## Abstract

Chest Ultrasound Scan (CUS) has been utilized in place of CXR in the diagnosis of adult pneumonia with similar or higher sensitivity and specificity to CXR. However, there is a paucity of data on the use of CUS for the diagnosis of childhood TB. This study aimed to determine the diagnostic accuracy of CUS for childhood TB. This cross-sectional study was conducted at the Mulago National Referral Hospital in Uganda. Eighty children up to 14 years of age with presumptive TB were enrolled. They all had CUS and CXR performed and interpreted independently by radiologists. The radiologist who performed the CXR was blinded to the CUS findings, and vice versa. Radiologists noted whether TB was likely or unlikely. A two-by-two table was developed to compare the absolute number of children as either TB likely or TB unlikely on CXR or CUS. This was used to calculate the sensitivity and specificity of CUS when screening for TB in children, with a correction to accommodate the use of CXR as a reference test. The sensitivity of CUS was 64% (95% CI 48.5%-77.3%), while its specificity was 42.7% (95% CI 25.5%-60.8%). Both the CUS and CXR found 29 children with a likelihood of TB, and 27 children unlikely to have TB. CUS met the sensitivity target set by the WHO TPP for Triage, and it had a sensitivity and specificity comparable to that of CXR.

## Introduction

Globally, of the estimated 1.1 million children who develop TB annually, only around 399,000 (36.5%) are notified to the National TB Programs (NTPs) [1]. The remaining patients are either not reported or never diagnosed. TB in children is underdiagnosed mainly because the signs and symptoms of TB in children, namely fever, cough, night sweats, weight loss or poor weight gain, visible neck swelling, and reduced activity, are not extremely specific and tend to overlap with other common pediatric conditions, such as pneumonia, HIV-associated lung disease, and malnutrition [2].

This is further complicated by the inability to confirm the disease, especially in younger children who cannot voluntarily expectorate sputum, which is the standard specimen used to

**Funding:** This study was not funded.

**Competing interests:** The authors have declared that no competing interests exist.

confirm the disease. Therefore, invasive methods, such as gastric aspiration and sputum induction, have been used [3]. When a sample is obtained, the paucibacillary nature (low bacterial load) of sputum in children compromises the diagnostic yield [4]. Hence, Chest X ray (CXR) is often used alongside clinical symptoms to make a presumptive diagnosis of Pulmonary Tuberculosis (PTB) in the absence of bacteriological confirmation [5].

Despite the proven utility of CXR, their hardware is expensive, with limited availability and accessibility in many low-resource settings with a high TB burden [6]. Owing to the scarcity of both equipment and skilled radiological staff to operate and interpret the images, the WHO recently recommended the use of Artificial Intelligence (AI)-assisted interpretation of radiographs, which would potentially reduce the requirement of skilled staff [7]. Despite this, however, AI is still underdeveloped in low-resource settings due to the high costs involved, and yet these settings have a high TB burden in children.

By contrast, Chest Ultrasound Scan (CUS) equipment is relatively inexpensive, portable, and accessible, does not use ionizing radiation, and may not require radiological staff [8, 9]. This is because health workers can be trained to perform basic point-of-care chest ultrasonography (POCUS) to aid in the diagnosis of TB. Ultrasound has been previously used to diagnose several diseases in low- and middle-income countries (LMICs) [10], notably with the focused assessment with sonography for HIV-associated extra-pulmonary TB (FASH) protocol [11, 12]. CUS has also been successfully used in the diagnosis of adult pneumonia with a previous meta-analysis suggesting that it had similar or higher sensitivity and specificity to CXR [13–15]. Some studies on the use of CUS for the diagnosis of childhood TB have been conducted in high-income settings [6]. However, there is a paucity of data on the use of CUS to diagnose TB among children in low-resource settings, which even have a higher TB burden when compared to high-income settings. The purpose of this study, therefore, was to determine the diagnostic accuracy of CUS in diagnosing TB in children using CXR as a reference test.

## Methods and methods

### Study design and settings

This cross-sectional study was conducted at the Acute Care Unit (Pediatric Medical Emergency Ward), Ward 16C (Pediatric Respiratory Ward), Ward 11 (Infectious Diseases and Neonatology Ward), and the Malnutrition Unit (*Mwanamugimu*) at Mulago National Referral Hospital in Kampala, Uganda, East Africa. This study was conducted between December 2020 and May 2021. These wards are the main diagnostic and treatment units for pediatric TB at the Mulago National Referral Hospital. On average, 12 children were diagnosed with TB (both pulmonary TB and extra-pulmonary TB) monthly [16]. These children were diagnosed based on the clinical signs and symptoms, with a review of a CXR if available, and Xpert MTB/RIF Ultra testing if a sputum sample could be obtained according to the national guidelines [17].

### Study population

This study was conducted among children aged 0–14 years with cough for more than 2weeks and the following symptoms: weight loss or failure to thrive over the last three months, persistent fever for >2 weeks not responding to anti-malarial treatment and having been in contact with someone with confirmed PTB. Children were excluded from the study if they were already on TB treatment or TB prophylaxis for > 72 hours. Respondents were enrolled using simple random sampling in preselected wards. All the study participants were de-identified and given unique study numbers.

## Sample size estimation

To determine the diagnostic accuracy of CUS in Pediatric TB. The formula for the sample size for diagnostic accuracy is as follows:

$$N = \frac{Z_{\alpha/2}^2 S_N (1 - S_N)}{d^2 x prevalence}$$

$Z_{\alpha/2} = 1.96$, the standard normal value corresponding to a 5% level of significance
$S_N = 0.76$, Assuming a sensitivity of Chest Ultrasound in the detection of TB of 76%.
$d = 0.1$, precision
$P = 0.89$, Prevalence of pulmonary TB among children under 15 [18]
We get a sample size of N = 80 participants.
A sample size of 80 participants was considered for this study.

## Measurements of variables

All study respondents had a CUS and a CXR. CXR was used as the reference test. They were independently interpreted by two radiologists, and if there was a discrepancy, a third radiologist was used as a tiebreaker. The radiologists who performed the chest ultrasound scans were blinded to the chest radiograph findings.

## Detecting TB using CUS

A portable, low-cost, greyscale ultrasound machine (Edan, Model U60) with a bandwidth of 5.0 to 10.0 MHz linear probes was used to scan the chest of the children in this study. The study radiologist was responsible for methodologically scanning all study children. The younger children were scanned from the mother or caretaker's lap and the older children were scanned in the sitting position to examine the lungs and pleura. The chest was divided into ten regions namely: left and right, supramammary, inframammary, lateral, suprascapular, and infrascapular. All regions were scanned in longitudinal and transverse (intercostal) planes. Mediastinal ultrasound was performed through the suprasternal notch. The child was placed in the supine position with the neck slightly extended to improve access to the suprasternal notch, and transverse and oblique views were obtained. All the cine clips were saved.

On CUS, the radiologist looked out for pleural effusions, an increased pleural gap, an interrupted pleural line, greater than 3-B lines, lymph nodes, consolidation, pericardial effusion, and any other relevant findings. Radiologists performing ultrasonography recorded the findings in a standardized form. Eventually noting if TB was likely or unlikely.

## Reference test-CXR

CXR was used as a reference test. To ensure the highest achievable quality of CXR reports, the posterior-anterior or anteroposterior views for younger children and lateral CXRs were independently reviewed by the study radiologists reporting on a standardized record sheet for consolidations, nodular opacities, hilar or paratracheal nodes, pleural effusion, and radiological diagnosis of TB likely or unlikely.

In cases of disagreement regarding the final diagnosis, a third radiologist was used as the tiebreaker. All the images were digitally achieved.

## Data analysis

Study characteristics were summarized using mean and standard deviation (SD) for normally distributed numerical data and median with interquartile range (IQR) for non-normally

distributed numerical data. Categorical data are summarized using frequencies and proportions.

A two-by-two table was developed to compare the absolute numbers of study participants who were identified as likely to have TB on both the CUS and CXR. Those unlikely to have TB on both CUS and CXR and those that were likely to have TB on CUS but not CXR, and vice versa. The sensitivity and specificity of CUS for diagnosing TB in children were calculated using CXR as a reference test. The sensitivity and specificity of ultrasound scans for diagnosing TB in children were corrected using the Staquet et al. correction method below [19]

$$\mathbf{CUS\ Sn} = \frac{eSnCXR + b(1-SpCXR)}{eSnCXR + f(1-SpCXR)}$$

$$\mathbf{CUS\ Sp} = \frac{c(1-SnCXR) + dSpCXR}{e(1-SnCXR) + fSpCXR}$$

Where **Sn** = Sensitivity
**Sp** = Specificity
And a, b, c, d, e, and f are defined in the table below.

|  | CXR |  |  |
|---|---|---|---|
| **CUS** | **TB Likely** | **TB Unlikely** | **Total** |
| **TB Likely** | a | b | a+b = g |
| **TB Unlikely** | c | d | c+d = h |
| **Total** | a+c = e | b+d = f | a+b+c+d = N |

## Ethical considerations

This study was approved by the Makerere University School of Medicine Research Ethics Committee (REC REF 2020–138). Informed parental/caregiver consent was obtained for all children below 8 years of age, while informed parental consent and child assent were obtained for all children above 8 years of age, in line with ethical research guidelines in Uganda. The study was conducted following the relevant guidelines and regulations of the Declaration of Helsinki. There was confidentiality regarding the medical information collected. The CXR and CUS investigations performed were already part of the routine standard of care.

## Results

We report our findings in accordance with 'The Standards for Reporting of Diagnostic Accuracy Studies" (STARD) statement [20].

## Patient characteristics

Table 1 presents the characteristics of the study participants. Eighty children underwent CUS and CXR examination (See Fig 1); of these, 42 (52.5%) were male. Their mean age in months was 42.9 (1–144). Most of the children (50.0% (40/80)) were between the ages of 1 and <5 years, with 21.3% (17/80) being less than one year. In contrast, 15.0% (12/80) and 13.7% (11/80) were aged 5–< 10 years and 10–15 years, respectively.

CUS found that 61.3% (49/80) of the children were likely to have TB with a mean age of 48.2 (2–144) months compared to 34.5 (1–144) months for those who were found unlikely to have TB on CUS. More females (57.1% (28/49)) were likely to have TB on CUS. Those unlikely to have TB on CUS were mostly male (67.7%, 21/31).

**Table 1. Characteristics of the study participants.**

| Characteristic | Total Participants (n = 80) | CXR findings | | CUS findings | |
| --- | --- | --- | --- | --- | --- |
| | | TB Likely (n = 33) | TB Unlikely PTB (n = 47) | TB Likely (n = 49) | TB Unlikely PTB (n = 31) |
| Sex, n (%) | | | | | |
| Male | 42 (52.5) | 16 (48.5) | 26 (61.9) | 21 (42.8) | 21 (67.7) |
| Female | 38 (47.5) | 17 (51.5) | 21 (44.7) | 28 (57.1) | 10 (32.3) |
| Mean Age in months | 42.89 | 39.82 | 45.04 | 48.22 | 34.45 |
| Age Ranges, n (%) | | | | | |
| < 1 year | 17 (21.3) | 7 (41.2) | 10 (58.8) | 8 (47.1) | 9 (52.9) |
| 1-<5 yrs. | 40 (50.0) | 17 (42.5) | 23 (57.5) | 25 (62.5) | 15 (37.5) |
| 5 yrs. -<10 yrs. | 12 (15.0) | 6 (50.0) | 6 (50.0) | 7 (58.3) | 5 (41.7) |
| 10 yrs.-<15 yrs. | 11 (13.7) | 3 (27.3) | 8 (72.7) | 9 (81.8) | 2 (18.2) |

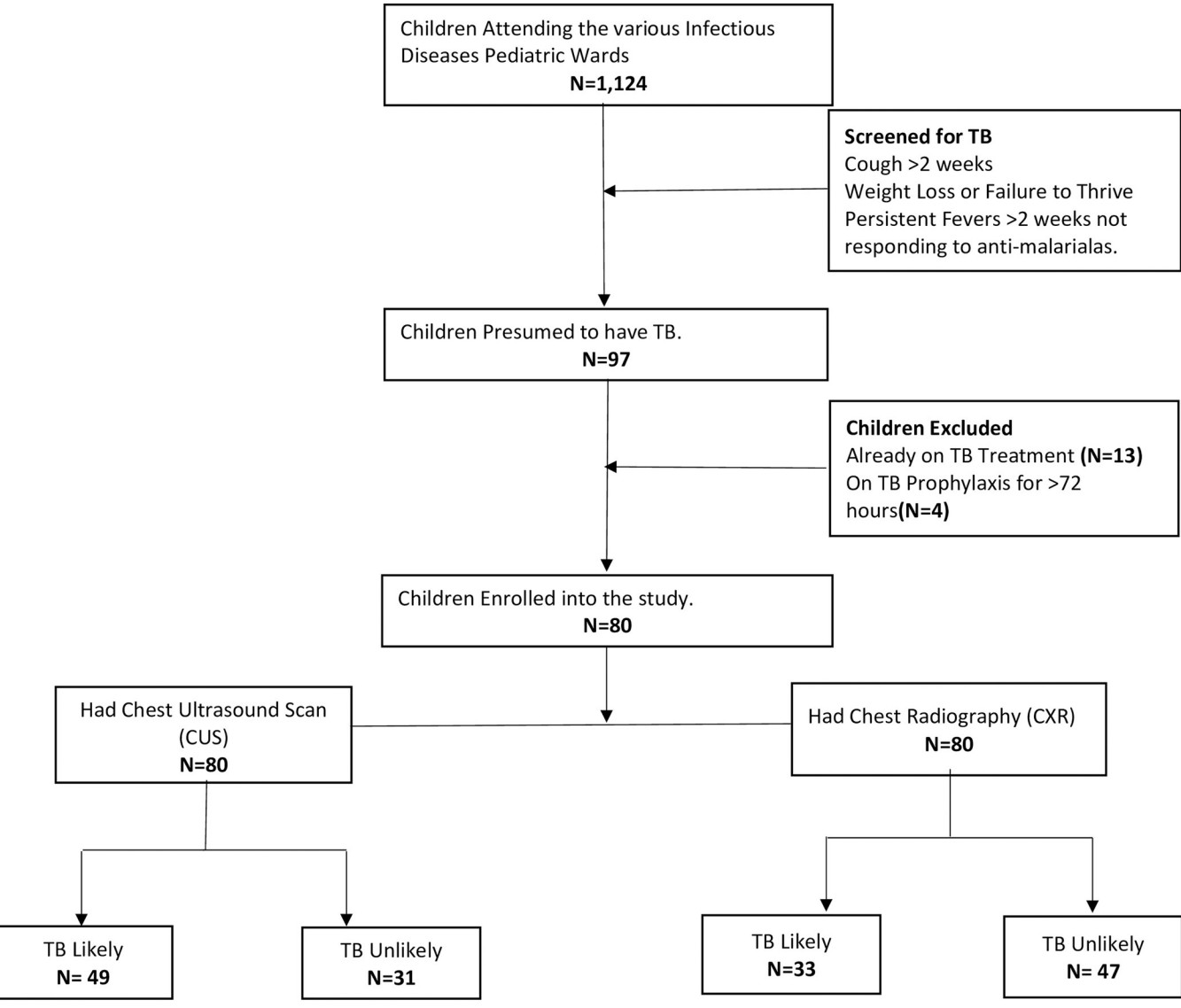

**Fig 1. Follow diagram for the diagnostic accuracy of chest ultrasound scan in diagnosis of childhood TB.**

**Table 2. Two by two table of chest ultrasound scan and chest radiography.**

|  | CXR | | |
|---|---|---|---|
| **CUS** | **TB Likely** | **TB Unlikely** | **Total** |
| **TB Likely** | 29 | 20 | 49 |
| **TB Unlikely** | 4 | 27 | 31 |
| **Total** | 33 | 47 | 80 |

On CXR, 41.3% (33/80) of patients were likely to have TB. Their mean age in months was 39.8 (3–144) compared to 45.0 (1–144) for those unlikely to have TB. Among those likely to have TB on CXR, there was an equal distribution among males (48.5% (16/33)) and females (51.5% (17/33)). Those unlikely to have TB on CXR were more likely to be male (61.9% (26/47)).

### Diagnostic accuracy of CUS in the diagnosis of childhood TB

Table 2 shows a two-by-two table with children who were either likely to have TB or unlikely to have TB on CUS or CXR. The CUS agreed with the CXR that twenty-nine children were likely to have TB while twenty-seven children were unlikely to have TB. However, there was a disagreement between the two modalities on 24 children. Twenty children were found likely to have TB on CUS but not on CXR. While the reverse was true for four children (CXR found them likely to have TB but not CUS). Using this information, we corrected for an imperfect reference standard. The sensitivity of CUS was 64% (95% CI 48.5%-77.3%). The specificity was 42.7% (95% CI 25.5% -60.8%).

### Discussion

This study aimed to determine the diagnostic accuracy of CUS when screening for Childhood TB. The sensitivity of CUS was 64% (95% CI 48.5%-77.3%), while its specificity was 42.7% (95% CI 25.5%-60.8%).

The CUS had a sensitivity of 64% (95% CI 48.5%-77.3%) and a specificity of 42.7% (95% CI 25.5% -60.8%). The sensitivity and specificity of CUS are comparable to those of CXR ((Sensitivity-67% & Specificity-48%) [21, 22]. Ultrasound scanning devices are relatively inexpensive, portable, more accessible, and do not result in exposure to ionizing radiation (6). We recommend screening for childhood TB using CUS as an alternative to CXR.

CUS meets the 66% (60%-77%) minimum requirement set for the sensitivity of a target product profile (TPP) for a triage tool. But its specificity (42.7%) falls short [23]. However, a CUS is a point-of-care test with a turnaround turn of 6 minutes compared to the 2.3 days for CXR [9]. We recommend adopting CUS as a triage or screening tool for TB, where those with suggestive features go ahead to have a confirmatory test. This is very important in low-resource settings where a CXR is not easily accessible, especially in remote areas, where ultrasound equipment is available. Thus, ultrasound is an invaluable initial investigative modality to aid patient management.

This study had several strengths. We used CXR as a reference standard, which is the most common imaging technique for the diagnosis of PTB, especially in low-resource settings, and few studies of a similar nature have been conducted from low-resource settings, yet these settings have a high burden of childhood TB. Thus, this study presents the diagnostic accuracy of CUS in this context. The CXR is not a gold standard for the diagnosis of PTB and has many limitations with wide interobserver variability in interpretation [9]. However, the use of CXR

may be useful to investigate which tool is better to use as a first-line imaging modality for the diagnosis of PTB.

In this study, it was observed that CUS had comparable sensitivity to CXR in diagnosing childhood TB. This study was conducted in a low-resource setting with a high burden of TB and HIV, a situation similar to many other low-resource settings in LMICs. Therefore, there is a need to adopt the routine use of CUS in such children as a form of screening investigation to identify children who may need further investigation or those who need immediate clinical management. The significance of CUS is amplified by the fact that ultrasound equipment is relatively accessible in many remote settings and many health workers have been trained to use the equipment. Therefore, using CUS in these children would not only quicken clinical management, but also reduce unnecessary expenditure for referring patients to tertiary hospitals, yet they would easily have been managed. Many LMICs can thus utilize the findings from this study to think of adopting the use of CUS as a preliminary investigation for children with presumed PTB.

Our study did have some limitations. CUS is supposed to be a point-of-care bedside diagnostic test; however, in our study, CUS was performed by a radiologist. This was mainly because there is limited expertise in performing ultrasound scans among clinicians, but the next logical step would be to train clinicians and compare their performance with that of radiologists. Indeed, in many countries, point-of-care chest ultrasound is performed by non-imaging healthcare workers.

In this study, we did not confirm the disease in children found to have TB on CUS. However, there is no perfect gold standard for the diagnosis of TB, Culture, which is supposed to be the "gold standard" despite having good specificity, has a sensitivity of only 60% (95% CI 46%-76%) [24]. Therefore, this study contributes to the validation process of using CUS in diagnosing childhood TB and provides a basis upon which many other studies can be conducted.

## Conclusion

The sensitivity and specificity of the CUS were 64% and 42.7%, respectively. CUS has a sensitivity and specificity comparable to that of CXR but has other advantages such as not exposing children to ionizing radiation. Unlike CXR, CUS can also be performed at the bedside by any attending health worker if trained. Since ultrasound is also widely available and relatively more accessible in low-resource settings, we recommend the use of CUS as a first-line imaging modality in children with presumptive TB in settings where access to other imaging techniques is limited.

## Supporting information

**S1 Checklist. STROBE statement—checklist of items that should be included in reports of observational studies.**
(DOCX)

## Acknowledgments

I would also like to thank my TEAM members for all the tremendous contribution towards this work. Our gratitude to the staff of the department of Radiology at Mulago National Referral hospital for offering the conducive environment to conduct this study. Our final gratitude goes to our research participants, the children without which this study would not have been possible together with their parents or caretakers.

## Author Contributions

**Conceptualization:** Geoffrey Erem, Caroline Otike, Faith Ameda, Aloysius Gonzaga Mubuuke, Michael Kakinda.

**Data curation:** Caroline Otike, Aloysius Gonzaga Mubuuke.

**Formal analysis:** Geoffrey Erem, Caroline Otike, Faith Ameda, Dorothy Irene Nalyweyiso, Aloysius Gonzaga Mubuuke, Michael Kakinda.

**Investigation:** Geoffrey Erem, Maxwell Okuja, Faith Ameda, Aloysius Gonzaga Mubuuke.

**Methodology:** Geoffrey Erem, Caroline Otike, Maxwell Okuja, Faith Ameda, Dorothy Irene Nalyweyiso, Aloysius Gonzaga Mubuuke, Michael Kakinda.

**Supervision:** Geoffrey Erem, Faith Ameda, Aloysius Gonzaga Mubuuke, Michael Kakinda.

**Validation:** Geoffrey Erem, Caroline Otike, Maxwell Okuja, Faith Ameda, Dorothy Irene Nalyweyiso, Aloysius Gonzaga Mubuuke, Michael Kakinda.

**Writing – original draft:** Geoffrey Erem, Caroline Otike, Faith Ameda, Dorothy Irene Nalyweyiso, Aloysius Gonzaga Mubuuke, Michael Kakinda.

**Writing – review & editing:** Geoffrey Erem, Caroline Otike, Maxwell Okuja, Faith Ameda, Dorothy Irene Nalyweyiso, Aloysius Gonzaga Mubuuke, Michael Kakinda.

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
