## [Decision Letter · Decision Letter 0]

17 Jul 2023

PONE-D-23-17231Diagnostic accuracy of chest ultrasound scan in the diagnosis of childhood tuberculosisPLOS ONE

Dear Dr. Erem,

Thank you for submitting your manuscript to PLOS ONE. After careful consideration, we feel that it has merit but does not fully meet PLOS ONE’s publication criteria as it currently stands. Therefore, we invite you to submit a revised version of the manuscript that addresses the points raised during the review process.

We look forward to receiving your revised manuscript.

Kind regards,

Mao-Shui Wang

Academic Editor

PLOS ONE

Journal Requirements:

   "This study was not funded"

Reviewers' comments:

Reviewer's Responses to Questions

**Comments to the Author**

1. Is the manuscript technically sound, and do the data support the conclusions?

Reviewer #1: Partly

Reviewer #2: Yes

2. Has the statistical analysis been performed appropriately and rigorously? 

Reviewer #1: Yes

Reviewer #2: Yes

3. Have the authors made all data underlying the findings in their manuscript fully available?

Reviewer #1: Yes

Reviewer #2: Yes

4. Is the manuscript presented in an intelligible fashion and written in standard English?

Reviewer #1: Yes

Reviewer #2: Yes

5. Review Comments to the Author

Reviewer #1: The authors describe the performance of point of care ultrasound for the diagnosis of pulmonary tuberculosis in children in a high burden country. Although not totally new, this data adds to the body of evidence for POCUS showing it is feasible and possibly replacing CXR in resource limited settings. However, neither CXR nor POCUS alone are able to diagnose TB in children as thee major manifestation is lymphadenopathy that is not specific for TB, therefore without bacterial confirmation of M.tb, there are some difficulties in calculating sensitivity of this diagnostic tool.

- Line 50: introduce alle introduction, e.g. CXR chest X-ray

- Line 67/68: add reference for studies

- There already exists a body of evidence for chest ultrasound for paediatric TB also from low-income countries. An overeview can be found: Ruby, L. C., Heuvelings, C. C., Grobusch, M. P., Andronikou, S., & Bélard, S. (2022). Transthoracic mediastinal ultrasound in childhood tuberculosis: A review. Paediatric respiratory reviews, 41, 40–48. https://doi.org/10.1016/j.prrv.2020.11.002: Pediatr Pulmonol

- Line 80: does this only refer to pulmonary TB or all TB manifestations? Please specify

- Line 101: Sentence is incomplete

- Table between line 146/147: insert caption, lowercase all letters

- Line 121: Please specify what criteria were used on ultrasound and CXR to classify findings as TB likely or TB unlikely (e.g. consolidation could also be pneumonia)

- Line 196-198 please reword sentence

- Line 200: a definite TB diagnosis cannot be made by CXR and clinical symptoms, and therefore a sensitivity or specificity for diagnosis of TB not be calculated unless further testing is available. How is the diagnosis of TB supported? Were any immunological tests performed? Was there any other evidence of M. tb from sputum/Stool/nasopharyngeal aspirate….? I suggest to compare pathological findings detected by POCUS versus CXR instead of sensitivities.

Reviewer #2: This is a well-constructed manuscript that gives insights into the use of CUS for TB and hopefully moves the field towards its integration into standard clinical care. The study had a clearly defined question, with assessments of the index and reference tests being independent and blinded. Furthermore, the tests were done on an appropriate spectrum of patients with all patients receiving the same reference test. The study used appropriate statistical methods and measures to report the study.

The data suggests that CUS is just as useful as CXR. However, my main concern is that they did not confirm the disease in children found to have TB on CUS.

Minor comments

Line 50: Define CXR at first use.

Line 79: pediatric tuberculosis should read pediatric TB to maintain consistence in the use of the abbreviation “TB”.

Line 85: 2weeks weeks – repeated word

Line 88: Pulmonary Tuberculosis (PTB). This is already defined on line 51.

Include a STARD checklist

6. PLOS authors have the option to publish the peer review history of their article (what does this mean?). If published, this will include your full peer review and any attached files.

Reviewer #1: **Yes: **Stephanie Thee PhD

Reviewer #2: No

---

## [Author Response · Author response to Decision Letter 0]

11 Aug 2023

Response to the Editors

We thank the editor for this comment. We have edited the manuscript to ensure that it meets the PLOS ONE style requirements. 

The authors received no specific funding for this work.

We thank the editor for this reminder. We have deleted the Ethics and approval and consent to participate under Declarations. Which were previously lines 255-262. 

Response to the Reviewers 

Response to Reviewer 1

The authors describe the performance of point of care ultrasound for the diagnosis of pulmonary tuberculosis in children in a high burden country. Although not totally new, this data adds to the body of evidence for POCUS showing it is feasible and possibly replacing CXR in resource limited settings. 

We thank the reviewer for this compliment. 

However, neither CXR nor POCUS alone are able to diagnose TB in children as thee major manifestation is lymphadenopathy that is not specific for TB, therefore without bacterial confirmation of M.tb, there are some difficulties in calculating sensitivity of this diagnostic tool.

We expected the reviewer to raise this comment. Bacterial confirmation for Childhood TB is an exception not the rule. Data from an operational setting has found that only 10% of childhood TB was confirmed using GeneXpert MTB/RIF Ultra and Culture is likely to miss up to half of the TB cases. Because of this there was a consensus agreement on a clinical case definition for classification of Intrathoracic Tuberculosis in Children. https://doi.org/10.1093%2Fcid%2Fciv581

The use of CXR or POCUS falls under Unconfirmed Tuberculosis using symptoms/signs suggestive of TB, and chest radiograph consistent with TB with both required to make a diagnosis of unconfirmed TB. We are trying to extend this to include findings on CUS. 

To find the diagnostic accuracy we would ideally need to compare the persons presumed to have TB on CUS compared to a Composite Reference Standard (CRS) given that there is no perfect reference standard for Childhood TB. Using other modalities that can diagnose TB like Culture, GeneXpert MTB/RIF Ultra, CXR and probably response to treatment. Using the AND/OR rule, so found with TB on Culture OR GeneXpert OR findings on CXR OR response to treatment. However, our study was not set up to use the CRS. Another possible alternative is the use of a correction method, as suggested by Staquet et al., which was deemed to be the most accurate in the literature. https://doi.org/10.1186/s12874-021-01255-4 Using CXR as a Reference Standard albeit an imperfect one, to get the diagnostic accuracy of CUS. 

- Line 50: Introduce an alle introduction, for example, CXR chest X-ray

We thank the reviewer for pointing this out. We have introduced CXR in full and the sentence now reads as follows...” Hence Chest X-ray (CXR) is often used alongside clinical symptoms to make presumptive diagnosis of Pulmonary Tuberculosis (PTB) in the absence of bacteriological confirmation” ...

- Line 67/68: add reference for studies

We have added a citation describing a systematic review of the diagnostic accuracy of point-of-care ultrasound for pulmonary tuberculosis. https://doi.org/10.1371/journal.pone.0251236. This includes all studies conducted on the subject for both children and adults. 

- There already exists a body of evidence for chest ultrasound for paediatric TB also from low-income countries. An overeview can be found: Ruby, L. C., Heuvelings, C. C., Grobusch, M. P., Andronikou, S., & Bélard, S. (2022). Transthoracic mediastinal ultrasound in childhood tuberculosis: A review. Paediatric respiratory reviews, 41, 40–48. https://doi.org/10.1016/j.prrv.2020.11.002: Pediatr Pulmonol

We have extensively reviewed the article suggested by the reviewer. Whilst we set out to find the diagnostic accuracy of Chest Ultrasound Scan for Childhood PTB, the suggested article was a review of transthoracic mediastinal ultrasound in childhood Tuberculosis. However, mediastinal lymph nodes could be pathognomonic of Tuberculosis in children. They could also be available in other diseases, and we used the availability of lymph nodes, consolidations, and effusions. 

- Line 80: does this only refer to pulmonary TB or all TB manifestations? Please specify

We have clarified whether it is PTB or both PTB and extra-pulmonary TB. The statement on line 80 now reads as follows

“On average, 12 children were diagnosed with TB (both pulmonary TB and extra-pulmonary TB) monthly’...

- Line 101: Sentence is incomplete.

We thank the reviewer for pointing this out. We have re-edited the sentence to read as follows...” A sample size of 80 participants was considered for this study” ...

- Table between line 146/147: insert caption, lowercase all letters

We have re-written the letters to all lowercase letters.

- Line 121: Please specify what criteria were used on ultrasound and CXR to classify findings as TB likely or TB unlikely (e.g. consolidation could also be pneumonia)

On CUS, the radiologists examined for pleural effusions, an increased pleural gap, an interrupted pleural line, greater than 3-B lines, lymph nodes, consolidation, and pericardial effusion. These were all suggestive of PTB but, the location, size and number of features are either suggestive of TB or not. Hence the classification by radiologists as either TB or not

While on CXR, the radiologist looked out for consolidations, nodular opacities, hilar or paratracheal nodes and pleural effusions. The size, location and combination of features were used by the radiologist to determine whether TB was likely or unlikely. 

- Line 196-198 please reword sentence

We have re-written line 196-198 to read as follows,” However, there was a disagreement between the two modalities on 24 children. Twenty children were found likely to have TB on CUS but not on CXR. While the reverse was true for four children (CXR found them likely to have TB but not CUS)” ....

- Line 200: a definite TB diagnosis cannot be made by CXR and clinical symptoms, and therefore a sensitivity or specificity for diagnosis of TB not be calculated unless further testing is available. How is the diagnosis of TB supported? Were any immunological tests performed? Was there any other evidence of M. tb from sputum/Stool/nasopharyngeal aspirate….? I suggest comparing pathological findings detected by POCUS versus CXR instead of sensitivities.

We thank the reviewer for this suggestion. We have a pre-print which compares the findings on https://doi.org/10.21203/rs.3.rs-2994488/v1. However, for this paper we set out to Diagnostic Test Accuracy of CUS for childhood TB. 

The processes and explanations have been provided above. 

Reviewer 2

Line 50: Define CXR at first use.

We thank the reviewer for pointing out this omission, the same omission was raised by Reviewer 1 and it was fixed.

Line 79: pediatric tuberculosis should read pediatric TB to maintain consistence in the use of the abbreviation “TB”.

We have made this change as suggested by the reviewer.

Line 85: 2weeks weeks – repeated word

We thank the reviewer for this observation. We have deleted the repeated word. 

Line 88: Pulmonary Tuberculosis (PTB). This is already defined on line 51.

We have used PTB, to ensure that there is consistency. 

Include a STARD checklist.

We thank the Reviewer for this comment. We have included having used the STARD Checklist in reporting our results. The statement reads as follows:” We report our findings in accordance with ‘The Standards for Reporting of Diagnostic Accuracy Studies” (STARD) statement [20]’’.

---

## [Decision Letter · Decision Letter 1]

1 Sep 2023

Diagnostic accuracy of chest ultrasound scan in the diagnosis of childhood tuberculosis

PONE-D-23-17231R1

Dear Dr. Erem,

We’re pleased to inform you that your manuscript has been judged scientifically suitable for publication and will be formally accepted for publication once it meets all outstanding technical requirements.

Kind regards,

Mao-Shui Wang

Academic Editor

PLOS ONE

Additional Editor Comments (optional):

Reviewers' comments:

Reviewer's Responses to Questions

**Comments to the Author**

1. If the authors have adequately addressed your comments raised in a previous round of review and you feel that this manuscript is now acceptable for publication, you may indicate that here to bypass the “Comments to the Author” section, enter your conflict of interest statement in the “Confidential to Editor” section, and submit your "Accept" recommendation.

Reviewer #2: All comments have been addressed

2. Is the manuscript technically sound, and do the data support the conclusions?

Reviewer #2: Yes

3. Has the statistical analysis been performed appropriately and rigorously? 

Reviewer #2: Yes

4. Have the authors made all data underlying the findings in their manuscript fully available?

Reviewer #2: Yes

5. Is the manuscript presented in an intelligible fashion and written in standard English?

Reviewer #2: Yes

6. Review Comments to the Author

Reviewer #2: (No Response)

7. PLOS authors have the option to publish the peer review history of their article (what does this mean?). If published, this will include your full peer review and any attached files.

Reviewer #2: **Yes: **Dr Humphrey Mulenga

---

## [Editor Report · Acceptance letter]

11 Sep 2023

PONE-D-23-17231R1 

Diagnostic accuracy of chest ultrasound scan in the diagnosis of childhood tuberculosis 

Dear Dr. Erem:

I'm pleased to inform you that your manuscript has been deemed suitable for publication in PLOS ONE. Congratulations! Your manuscript is now with our production department. 

Kind regards, 

on behalf of

Dr. Mao-Shui Wang 

Academic Editor

PLOS ONE